# Social Acceptance of Aquaculture in Spain: An Instrument to Achieve Sustainability for Society

**DOI:** 10.3390/ijerph17186628

**Published:** 2020-09-11

**Authors:** José Ruiz-Chico, José M. Biedma-Ferrer, Antonio R. Peña-Sánchez, Mercedes Jiménez-García

**Affiliations:** 1INDESS (University Institute for Sustainable Social Development), University of Cadiz, 11406 Jerez de la Frontera, Spain; josemaria.biedma@uca.es (J.M.B.-F.); rafael.pena@uca.es (A.R.P.-S.); mercedes.jimenezgarcia@uca.es (M.J.-G.); 2Department of General Economy, Faculty of Social Sciences and Communication, University of Cadiz, Avda. de la Universidad, 4-11406 Jerez de la Frontera (Cadiz), Spain; 3Department of Business Administration, Faculty of Social Sciences and Communication, University of Cadiz, Avda. de la Universidad, 4-11406 Jerez de la Frontera (Cadiz), Spain

**Keywords:** economy, society, environment, health, fish, quality, employment, pollution, consumer, public administrations

## Abstract

Aquaculture is a technique to produce food that is under debate, due to its possible consequences for altering the economy, traditional fishing included, or the environment, even with doubts about the health of consumers. This document studies its social acceptance from the point of view of carrying capacity. This term is defined as the level at which this activity begins to be disproportionate and poses important disadvantages for society. In this context, we conducted 803 surveys in six coastal provinces in Spain. The results show that the acceptance of these products is good, implying that aquaculture is far from reaching its saturation point in society. Additionally, the respondents gave a higher priority to socio-economic objectives than to environmental ones. We can conclude that the further development of this sector is advisable in these provinces. The general perception of aquaculture is better among men, and also among higher-income consumers. Informative activities should be organized to target these more hesitant groups. Production structures should be revised to overcome biases in the population about the idea that the food obtained from aquaculture harms the environment or is less natural or healthy. The possible abuse of feed and chemicals spreads this idea, and this could affect the taste and quality adversely.

## 1. Introduction

Spain is one of the most important fishing European countries, being very relevant in terms of production, employment, fleet, consumption of fish, and aquaculture [1,2]. This technique is an important supply of quality products. However, it tends to be known primarily for producing sea bream and sea bass, although this is only part of it. Aquaculture can be defined as the farming of aquatic plants and animals, mainly fish, crustaceans, and molluscs, as an obvious evolution of traditional fishing activities [3]. In recent times, this sector is a significant business producing food, pharmaceutical and industrial materials, and storage for restocking or ornamental goals, generating 12 million jobs worldwide [4].

Spanish aquaculture is the largest one in the European Union [4]. This activity produced in 2016 283,831 tonnes, well above the United Kingdom (194,492 tonnes) and France (166,640 tonnes). It is valued at €449.4 million, with 17,811 employees. The main Spanish species obtained are blue mussels (215,855 tonnes), sea bass (23,445 tonnes), rainbow trout (17,732 tonnes), and sea bream (13,740 tonnes). Moreover, aquaculture has all the requirements to be studied as a strategic sector [5].

There has been a revolution in this activity in recent years [6,7]. These changes are due to major advances in science and technology, better degrees of development in species that generate considerable economic benefits, and improved trade performances. [3]. However, aquaculture is a production technique which is not above discussion but is becoming more efficient [7]. Now, the same area can be used to get 10 times more production than agriculture. The conversion rate and production index of certain foods has even tripled. This sector is also a leading commercial business in certain areas, particularly in less developed countries, thus promoting specialization in some species.

Other advantages of aquaculture are, for example, its potential to reduce the over-exploitation of resources in its aquatic environment, and to ensure a supply for a larger population of such products that are increasingly consumed [8]. Its production can supply quality products in a controlled way, with the possibility of partial harvests and regular delivery to markets [9]. This method offers economically adequate and sustainable resources for producers. Aquaculture can also be developed in aquatic regions to populate or restock them for commercial or environmental purposes (native and exotic species, fishing, or sports).

On the other hand, some studies have highlighted the potential dangers of environmental damage and the fact that some products in aquaculture could negatively affect the final consumer [10]. Other works focused on pollution, a problem usually dealt with from the perspective of social pressure and also for its own necessity [8,11]. We should take into account that aquaculture is carried out in a barrier-free aquatic zone, and these conditions will affect the growing organisms. This concern for the environment supports its survival.

Aquaculture is also a competing sector for resources and the quality of the environment with other nearby activities [11]. Additionally, aquaculture also involves socio-economic factors, some of them requiring specialised regulations directly. Therefore, there are frictions with agents such as ports, nature, traditional fishing, industry, housing development, and tourism. Indigenous species can also be affected in certain areas. The use of chemicals may also harm the quality and taste of the food.

In our case, the main objective of our study was to study the social acceptance of aquaculture in the regions in which it is practised to help determine its relative level that is considered “acceptable” for its development. As a result, we can apply this information about acceptance, and the factors to which the respondents are found to be more or less sensitive, to check the current situation in these provinces compared to its social carrying capacity. This study is very innovative as not much research has targeted this critical area, and specifically in Spain. Our conclusions can address new ways of work in the future with interesting repercussions on aspects such as society, economy, production, or environment.

We can define carrying capacity as the intensity of a practice in a given environment that can be sustained indefinitely over time depending on the availability of resources and external pressures [12]. There are various types of carrying capacity: physical (a geographical area with adequate conditions for specific species and methods), productive (the density of factors required to maximize production), ecological (the maximum density of individuals with which acceptable changes are produced in the ecosystem), economic (the amount of production that leads to tolerable modifications in the main economic activities of a specific area, or also the production that can be absorbed by the market without problems), and social (the amount of production suitable to society—businesses, residents, and environmental conservation organizations). In other words, this last capacity would be the point at which other social uses begin to be excessive because of the development of this technique [13]. In the case of aquaculture, it would not be adequate if this growth led to adverse consequences on the environment or the gross domestic product of an area derived from other activities such as tourism.

Limited studies have tried to analyze the indicators of socio-economic sustainability in aquaculture across a series of countries. These findings revealed a gap in the current literature and an opportunity to develop empirical research, such as that described in this article.

There have been previous works focused on social carrying capacity from a wide perspective. For instance, Prato focused on protected ecosystems [14]. Some papers have addressed the topic of tourism [15,16,17,18,19,20]. A more recent study has analyzed this concept from an economic standpoint strictly [21].

Some other studies have addressed the general concept of marine issues [22]. Some of them targeted the biological standpoint [23]. With regard to aquaculture and its socio-economic perspective, some works have defined indicators and critical limits to measure social carrying capacity [24]. They have pointed out that this must be as broadly representative of the groups involved and should be supervised over time and in similar regions where aquaculture has not been developed.

There are several works specifically about sectors such as oysters [25], bivalve molluscs [12,13,26,27,28], or coral reefs [29], emphasising environmental factors [11,30] or following a socio-economic perspective to analyze inland and coastal alternatives [31]. A work about social carrying capacity on some islands in Greece noted greater support among the inhabitants than the tourists [32]. Environmental approaches were highlighted in several studies [33], detecting that if the product is obtained in a more eco-friendly way, there is a willingness to pay higher prices for it—for example, for the case of salmon in Scotland [34].

In 2006, from an ecosystem perspective, FAO (Food and Agriculture Organization of the United Nations) put forward recommendations for aquaculture management. They must take into account all its services and purposes without posing risks to society. This proposition also pointed out that other departments, policies, and common goals must be considered to improve the welfare and equity of the affected population about this topic.

In this context, the objective of this work is to analyse the acceptance of aquaculture products from the perspective of social carrying capacity, specifically in some geographic areas where this sector is established. This study tries to detect if there are development possibilities for this activity from a socio-economic point of view. We can see if the population accepts this technique and its benefits (employment, cheaper prices, products obtained more regularly during the year, etc.). However, the respondents can reject it due to the existence of some doubts about possible negative externalities in the region (chemical feed, pollution, alteration of the natural environment, threats to other sectors, etc.). For this reason, we aim to detect if this sector should increase its activity in the future, which would indicate that it has not reached its saturation point.

Adopting this approach, after this introduction we describe the methodology employed in our research in the next section. Then, we present the results of the fieldwork carried out. After that, we develop the discussion, comparing them with other works, and expose the main conclusions in the final sections of our work.

## 2. Materials and Methods

As explained above, the general objective of determining the relative position of the carrying capacity of aquaculture is to promote its development, avoiding unacceptable alterations in the ecosystem and socio-economic structures simultaneously. To this end, we first consulted secondary sources to review the theoretical framework of this issue. Then, this material was used as a documentary basis for drafting the questionnaire. The primary sources used to design the survey will provide a large amount of information for analysis by supplying direct evidence from the participants in the study in Spain.

The questionnaire, which was structured, had several parts where the respondents were asked about their consumption and buying behaviours, level of knowledge about this topic, opinions about aquaculture compared with traditional fishing, and the impact on the environment from different perspectives. The questionnaire contained a wide typology of questions: open and closed-ended, single- and multiple-choice answers, Likert scales, and comparisons in pairs. All of them had passed the convenient pre-tests. It ended with demographic questions (gender, place of residence, income, etc.).

The fieldwork was then planned using a sample framework prepared exclusively for this purpose. A polling company conducted face-to-face interviews in shopping centres in western Andalusia (Cadiz and Huelva) and the Mediterranean area (Murcia, Alicante, Castellon, and Tarragona). Aquaculture has been developed in these provinces and represented the geographical framework for our analysis, because they worked with different collection methods (cages and estuaries).

The survey was targeted at adults, using random sampling and set provincial and Nielsen geographical area classifications. The company collected the data in September 2018. Initially, our objective was to obtain 800 final surveys. This company verified 20% of the respondents at least, ratifying that the person concerned has actually been interviewed and also checking the consistency of the answers. After that, the company deleted those cases that did not pass this test.

The 844 surveys received were reduced to 803 finally after removing one more time those that did not pass the researchers’ filter before definitive analysis. This supposed a sampling error of ± 3.53% with a confidence level of 95.5%, under the hypothesis of maximum uncertainty in proportions (*p* = *q* = 0.5).

Table 1 presents the overall description of the sample distribution, without considering other possible quotas.

Finally, we subjected the data obtained to various statistical analyses. First, we carried out univariate analyses to determine the position and dispersion measurements, and also frequency distributions. Bivariate analyses were also conducted to identify potential dependencies between variables with Pearson’s X^2^ test or representativeness measurements of the means for subgroups by means of Snedecor’s F-distribution.

## 3. Results

Table 2 describes fish consumption by the respondents, obtained from aquaculture, at home or away. More than one third of the sample confirmed that the fish consumed at home was obtained from aquaculture, in comparison with almost one third who said it was not. These percentages were higher for the case of consuming it outside (hospitality services, for example); more than half of these respondents were unaware of its provenance.

By gender, the percentage of male consumers of this fish at home (40.71%) was higher than women (34.06%). The level of this consumption increased with income directly, reaching from 27.81% for the respondents with lower incomes to 46.34% for those above €2400 (*p* = 0.0515 (X^2^)). In the case of consuming it away from home, the difference is not so large according to these respective classifications.

Table 3 presents a good satisfaction with this kind of food, as 43.21% of respondents admitted that they were very or quite satisfied with it. However, almost one fifth stated little or no satisfaction at all. In the case of the study by gender, women were slightly more polarized than men were, with a greater proportion of “very satisfied” (9.96% of women vs. 7.92% of men), as well as “not at all satisfied” (11.04% vs. 10.26%). Moreover, according to the income level, the groups “very and quite satisfied” improve from people with the lowest incomes (28.99%) to those earning more than €2400 (54.77%). On the contrary, consumers with less satisfaction are those with incomes below €900 (17.16%), a percentage that decreases to 7.14% for people with higher incomes. The possible rejection, therefore, is lower for people with higher purchasing power.

Based on an open question, Table 4 presents the way the respondents valued the different advantages that aquaculture supposed to society. These percentages highlighted an improvement in their economies, cheaper prices, and more variety and quantity of fish all year long. In general, men and women see it similarly. In detail, women perceive it as a healthier fish (12.77%) than men (11.14%) do, while the opposite occurs with the possible respect for the environment (1.52% of women and 3.52% of men).

Regarding the income level, the respondents with higher incomes were the most receptive to aquaculture, being the most interested in the affordable price (30.95%), possible economic improvement (52.38%), and respect for the environment (4.76%), but least valued the variety and quantity of products (14.29%). These opinions are precisely the opposite for the lower-income respondents (23.67%, 27.81%, 0.59%, and 15.98%, respectively). In the same way, the possibility that aquaculture does not represent any advantage falls from 18.34% for those with an income below €900 to only 4.36% for those above €2400. On the other hand, the ones with the lowest incomes are more reluctant, highlighting a minority the possible respect for the environment (0.59%) and being also those who are more unaware of its advantages (14.79%) (*p* = 0.0019 (X^2^)).

Concerning the disadvantages associated with aquaculture companies, Table 4 also shows the opinions obtained from another open question. Half of the respondents considered that this sector did not suppose any disadvantages in terms of activity. Data show the perception of fish being unnatural or unhealthy as a consequence of the abuse of chemicals in this industry. Both men (20.23%) and women (14.50%) highlighted this opinion. The respondents also remarked the probability of worse taste and quality (9.38% and 6.93% for men and women, respectively). Women were more receptive to environmental effects (4.11%) than men (2.35%) were, although there were no relevant differences between men and women regarding the adverse effects on traditional fishing (6.74% and 6.49%) (*p* = 0.0051 (X^2^)).

By the income level, those respondents with less than €900 were the least concerned about its potential damage to the environment (2.37%) and the abuse of chemicals and feed (11.24%). In contrast, those with more than €2400 are the most worried with the environment (4.76%), its possible damage to traditional fishing (21.43%), and its poorer quality or flavor (11.90%). The respondents with higher incomes are more aware of the drawbacks of this technique. In fact, this group cites the highest number of disadvantages (just 14.29% in “No answer”), while it considers least that it does not produce any (42.86%).

Table 5 presents comparisons in pairs of possible objectives about aquaculture management to determine which pair of them was the most important to the respondents and to propose a final order for it. With a 17-point scale, a minimum value of 1 point would show the highest level of support for the first objective, and a maximum of 17 would involve the highest one for the second one. The core rating of 9 points would reflect a certain indifference between both endpoints. Table 5 balances in pairs three socio-economic goals (disaggregating it in employment, fish quality, and increase in wealth) and environmental ones (with pollution, visual impact, and effects on nature in this case). This table shows that the first group of objectives received a higher priority (8.5827 points).

Analyzed by gender, socio-economic goals were more relevant to men (7.9941), while women were more indifferent (9.0197) (F-distribution *p* = 0.0093). In the case of income level, this objective was prevalent across all categories, particularly for the respondents with an income exceeding €2400. In fact, the valuation of environmental objectives increases as income does.

In short, when considering socio-economic objectives, the priority was “employment”, “fish quality”, and “increase in wealth”. In the case of the environmental group, the order was “pollution”, “effects on nature”, and “visual impact”. We observe a similar dispersion in all comparisons, indicative of some degree of homogeneity, although the level of disparity was slightly greater in the case of “employment vs. fish quality”.

Characterizing the subgroups involved in this analysis yielded the following specific observations; the order of preference was the same for both men and women, although women were more hesitant to choose between the two types of objectives. We did not identify significant differences in the case of income level.

Table 6 presents the results of how aquaculture should develop in the respondents’ region of residence in the coming years. Almost half of them thought that this sector should develop its activity. Instead, one tenth, the most reluctant ones, thought that it should be lower.

After classifying these data by gender, we saw that men were more positive with this sector, because 50.73% stated that this activity should be increased, as opposed to women (41.34%) (*p* = 0.0402 (X^2^)). By income level, the opinion about support for aquaculture is the majority in all groups and grows progressively from 37.87% for those with lower incomes to 50% for the respondents with more than €2400. Otherwise, the option to reduce this activity decreases irregularly from 11.83% for the lowest incomes to 4.76% for those consumers with the most purchasing power.

Table 7 compares aquaculture and traditional fishing according to certain factors, ranked by its level of acceptance. In detail, the respondents considered that aquaculture had more affordable prices and provided a greater quantity of fish to the markets. Instead, they believed that traditional fishing offered more variety and a better quality of products, being healthier and also more environmentally friendly.

These responses can be categorized according to the predetermined sample subgroups with a comparable model, as detected previously. By gender, men were more receptive to aquaculture, considering it more affordable than traditional fish (74.78%) than women did (69.70%). Men perceived also that it generated more employment (44.87%) and was of higher quality (45.16%) than women did (38.31% and 36.36%, respectively).

The perception of aquaculture fish as being better rose with the income level. Especially, the rating of the price as cheaper increased from 60.95% for lower incomes to 78.57% for higher levels (*p* = 0.0010 (X^2^)). The same profile presents the analysis of availability in supermarkets, increasing from 52.66% to 73.81% (*p* = 0.0256 (X^2^)).

In contrast, all the groups considered traditional fishing more eco-friendly and healthier in all categories, but this decreased as income increased. In the first case, the support goes down from 63.31% for incomes below €900 to 52.38% for those above €2400. The same occurs in the case of the perception of healthier fish, falling from 82.25% to 59.52% between both categories (*p* = 0.0388 (X^2^)), its variety (from 78.70% to 57.14%—*p* = 0.0032 (X^2^)), its taste (from 87.57% to 69.05%—*p* = 0.0277 (X^2^)), or its quality (from 88.17% to 78.57%).

Between both positions, the group with lower incomes was the only one who stated that traditional fishing created more employment (47.93%); the other respondents chose aquaculture on an increasing basis up to 52.38% for the highest income levels (*p* = 0.0440 (X^2^)). A comparable behavior was clear in the case of the quality of employment. Incomes below €900 and between €900 and €1200 opt for traditional fishing (39.05% and 39.84%, respectively) to increase the preference for aquaculture up to 54.76% in the case of higher incomes (*p* = 0.0039 (X^2^)).

Table 8 presents the percentage of respondents who thought that public administrations supported this kind of fishing. One third specified that aquaculture was supported to a greater extent. Analyzed by gender, this support was higher in men (37.54%) than in women (30.09%) across all options (*p* = 0.0012 (X^2^)).

Finally, concerning income, levels of backing for aquaculture regularly rose with the income level, from 25.44% for the lowest incomes to 42.86% at the higher ones. We can remark some stability for the rest of possible answers regarding this variable, highlighting the decrease in undecided responses from 31.95% for incomes below €900 to 16.67% for those above €2400. This could indicate that respondents have a clearer opinion when they have a higher level of income.

## 4. Discussion

In this work, we aimed to analyze the level of acceptance of aquaculture activity by the inhabitants in certain Spanish regions where this sector is located. From the perspective of social carrying capacity, we aimed to study the possibility of a further development compared with its saturation point. We can define it as the limit beyond which it would lead to negative consequences for the ecosystem and social structures in its area. In this sense, we tried to identify potential rejections in the population, based for example on alterations in the environment, the use of possible chemical products that could affect the health of consumers, or threats to traditional sectors due to competitiveness for the resources. This approach has not been studied in depth, offering a broad range of possibilities for further research.

Based on the results obtained, as the main point of the study, a substantial proportion of respondents consumed and were satisfied with aquaculture fish (43.21%), while who are not at all only account for 10.71%. This implies a low level of rejection for this type of fish and a way of improvement for this sector at the same time. Furthermore, there was a general view that this sector should increase its activity in the near future (45.33%) [35]. These data imply that Spanish aquaculture is far from its saturation point, represented by the industry’s maximum social carrying capacity. Further development of the sector can be recommended without causing serious problems for its social environment, being even advisable for society.

In this context, the Spanish population knows aquaculture fish and consumes it regularly [4]. It means that they know to some extent the nature of the problem we are studying. This analysis showed that 36.74% of aquaculture fish was consumed at home, but only 22.79% did it away. In fact, more than half of the respondents ignored the source of fish in this second case. These percentages are similar, or a little higher, to the consumption habits in the European Union [4]. In recent times, traceability is necessary, and it is recommendable to offer this information in services such as hospitality, mainly in the case of healthy food and high-quality products.

Consumers are very familiar with this kind of fish [4]. The population knows the advantages that aquaculture brings to their areas [36], reinforcing the social acceptability of the sector, as one of the main indicators and critical limits to measure its social carrying capacity [24]. Data emphasize that the respondents agree that aquaculture offers cheaper prices and more quantity and variety of products, in the same line as Claret et al. [37]. It is also perceived as healthy, but not very eco-friendly as it was supposed.

According to its possible disadvantages, we must highlight that half of the respondents (50.19%) did not express any specifically for this technique. This is important when identifying the possible rejections of aquaculture. They would bring the sector closer to its maximum carrying capacity, marking the way forward to guarantee its future sustainability. In this manner, companies should control more strictly the inappropriate use of chemicals and artificial feed (16.94%) and look after the taste and fish quality (7.97%), because they can affect respondents’ perception of healthy food. These efforts should be made both from the technical and commercial perspectives to improve the consumer’s acceptance.

Social carrying capacity was previously defined as the level of production acceptable to society, mixing social and environmental aspects [12]. It is important to know the priorities that consumers have about these elements. In this sense, when the respondents assessed several pairs of objectives related to aquaculture, we noted that they gave a higher relevance to socio-economic benefits (the final priority was “employment”, “fish quality”, and “increase in wealth”) over environmental factors (where “pollution”, “effects on nature”, and “visual impact” were prioritized). Anyway, these groups of objectives should not be ignored if the goal is to reinforce the strategic position of this sector. These strategies should be strengthened so that customers perceive these businesses as environmentally responsible institutions because, for example, the valuation of environmental objectives increases as income does.

These aspects are partially consistent with previous studies [34] that focused on environmental factors associated with salmon aquaculture in Scotland. Certain previous works have already outlined environmental problems and health threats related to this kind of activity [38]. Other research has pointed out that consumers could accept more expensive food if better safety was guaranteed [39,40]. These factors were also relevant to our current study, although they ranked them behind the socio-economic objectives. We should also consider that this last paper does not focus exclusively on environmental factors, or the case of salmon specifically.

Concerning the support of public administrations, the third part of the respondents (33.25%) considered that aquaculture was more backed. It is essential to avoid consumers seeing this sector as a menace to traditional fishing if it receives more favorable support from public administrations. It could cause a possible rejection towards aquaculture which could negatively affect its social carrying capacity.

Finally, we would like to remark about our sampling profiles that the acceptance of fish consumed has been previously studied from these standpoints (gender, income level), but none of them focused on aquaculture. We detected two groups which are particularly reluctant to use aquaculture: women and people with lower incomes. They are more aware of its disadvantages and opt for traditional fishing to a greater extent. This is a useful contribution of our work to this field of knowledge and the main reason why we consider our research innovative.

## 5. Conclusions

We can conclude that there is a good level of acceptance of aquaculture fish in these Spanish provinces, which would place this technique far from its saturation point. The respondents knew about it, consumed it frequently, and were aware of its advantages and disadvantages. Moreover, almost half of them considered that this sector should increase its activity in the coming years. Consequently, the further development of the sector in this area can be recommended from a social point of view.

Anyway, as a possible means of improvement, although consumers perceived aquaculture fish as healthy, they were very concerned about the possible use of chemicals and their effect on quality and taste. Furthermore, some respondents considered this sector as a menace to traditional fishing, even receiving more support from public management. Conversely, they did not consider aquaculture companies as very environmentally responsible. These aspects are very important to improve the competitive position of aquaculture in these regions. As practical implications of this work, we can recommend the diffusion of key information about ingredients used in production, socially responsible attitudes, or environmental policies. This would help to control possible rejections towards this sector, thus guaranteeing better social sustainability in the future.

Among the different profiles of the respondents, the perception about aquaculture was better in men than in women, who opted for traditional fishing. Last, the acceptance of this technique improved as income rose. The respondents with higher incomes were more in favor of aquaculture, knew it better, and were more conscious of its advantages and disadvantages. However, the respondents with a lower income usually had less confidence in these products, being more reluctant and preferring traditional fishing. Accordingly, these critical profiles become the main collectives for the sustainable development of this activity. Campaigns (information, advertising) should be focused on them to emphasize the advantages this technique provides its region.

At this point, our study had some limitations that we must make clear. These could be improved in the coming years, representing many interesting future studies. Firstly, our interest in socio-economic aspects made us follow a more quantitative methodology, more common in social sciences. If we conduct qualitative or experimental research, our data may be enhanced. We can also apply more analytical methodologies to confirm our findings. Such methods could be used in successive studies.

In the future, moreover, our analysis could be focused on a specific province, extended to other regions not included in this work, such as northern Spain or other different countries. In addition, we can also develop to see if the acceptance of aquaculture by the respondents is different towards local or foreign production.

Finally, the last factor to consider is the age of the respondent. This factor was not considered in this study because the polling company excluded it when it was first investigated. It may also be interesting to focus on the level of education or previous knowledge of the respondents as a variable related to their income. Qualitative methodologies could be useful to detect favorable attitudes or prejudices in this regard. Both factors might represent interesting future lines of research.

## Figures and Tables

**Table 1 ijerph-17-06628-t001:** Sample distribution.

Gender	n	%	Income ^1^ (€)	n	%
Women	462	57.53	<901	169	21.05
Men	341	42.47	901–1200	251	31.26
			1201–1800	219	27.27
			1801–2400	122	15.19
			>2400	42	5.23
Total	803	100.00	Total	803	100.00

^1^ Gross monthly figures.

**Table 2 ijerph-17-06628-t002:** Aquaculture fish consumed by the respondents.

	At Home		Away from Home	
	n	%	n	%
I consume aquaculture fish	295	36.74	183	22.79
I do not know if the fish I consume come from aquaculture	251	31.26	420	52.30
I do not consume aquaculture fish	225	28.02	129	16.06
I do not consume fish	32	3.99	71	8.84
Total	803	100.00	803	100.00

**Table 3 ijerph-17-06628-t003:** Satisfaction with aquaculture fish.

	n	%
Very	73	9.09
Quite	274	34.12
Somewhat	171	21.30
Not very	71	8.84
Not at all	86	10.71
No answer	128	15.94
Total	803	100.00

**Table 4 ijerph-17-06628-t004:** Advantages and disadvantages of aquaculture.

Advantages	n	%	Disadvantages	n	%
Improvement in economy and employment	301	37.48	Abuse of chemicals and feed	136	16.94
Cheaper prices	195	24.28	Poorer quality and taste	64	7.97
More variety and quantity	164	20.42	Damage to traditional fishing	53	6.60
Healthy. Better quality.	97	12.08	Damage to environment	27	3.36
Eco-friendly	19	2.37			
Others	15	1.87	Others	23	2.86
None	88	10.96	None	403	50.19
No answer	90	11.21	No answer	135	16.81
Total	969	120.67	Total	841	104.73

**Table 5 ijerph-17-06628-t005:** Comparison of objectives.

	Mean	St. Dev ^1^
Socio-economic vs. environmental	8.5827	5.5133
Employment vs. fish quality	8.0113	5.6370
Employment vs. increase in wealth	7.3529	5.4413
Fish quality vs. increase in wealth	6.4198	4.9648
Pollution vs. visual impact	5.9849	4.8884
Pollution vs. effects on nature	7.1777	5.1951
Visual impact vs. effects on nature	10.1069	4.9465

^1^ St. Dev. = Standard deviation.

**Table 6 ijerph-17-06628-t006:** Opinions of the respondents about “Aquaculture should in the near future”.

	n	%
Increase	364	45.33
Remain the same	236	29.39
Decrease	82	10.21
No answer	121	15.07
TOTAL	803	100.00

**Table 7 ijerph-17-06628-t007:** Comparison of fish obtained from aquaculture/traditional fishing.

	Aquaculture		Traditional		No Answer	
	n	%	n	%	n	%
Cheaper prices	577	71.86	114	14.20	112	13.95
More supply in supermarkets	524	65.26	129	16.06	150	18.68
More employment	330	41.10	309	38.48	164	20.42
Better quality employment	322	40.10	266	33.13	215	26.77
More eco-friendly	181	22.54	452	56.29	170	21.17
More variety	175	21.79	545	67.87	83	10.34
Healthier	105	13.08	596	74.22	102	12.70
Better quality	69	8.59	649	80.82	85	10.59
Better taste	48	5.98	646	80.45	109	13.57

**Table 8 ijerph-17-06628-t008:** Perception of support from governments.

	n	%
Traditional fishing	116	14.45
Aquaculture fish	267	33.25
Both of them	93	11.58
Neither	134	16.69
No answer	193	24.03
TOTAL	803	100.00

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
