# Peer review of "Social Acceptance of Aquaculture in Spain: An Instrument to Achieve Sustainability for Society"

_ijerph, 2020, doi:10.3390/ijerph17186628_

Round 1
Reviewer 1 Report
This submitted paper investigates the social acceptance of aquaculture using carrying capacity as an analytical framework. Carrying capacity refers to the point at which the presence of aquaculture tends to be excessive and generate difficulties in the society. To this end, the authors conduct 803 face-to-face surveys in six coastal provinces in Spain. Their main finding suggests a high level of acceptance for aquaculture products. They conclude that aquaculture is far from its social saturation point.
This submission is very interesting. It tackles in an original way an important question concerning the economy, health and the environment. This paper seems to me publishable in the journal even if I suggest a minor revision.
First, it would be useful to pay more attention to the differences in responses across income categories. More specifically, I suggest that the authors give more economic insights into these differences. For example, is there an unequivocal link between income level and price sensitivity? How to explain the fact that "respondents with higher levels of purchasing power were the most 193 interested in the affordable price" (page 5)?
Second, is the surveyed population representative of the inhabitants of the different regions studied?
Third, respond with lower income tends to be less concerned about the possible environmental damage of aquaculture. Is it due to a lack of expertise or to other factors?
Last, for future works, it could be interesting to assess to what extent the sensitivity of the respondents differ according to the local aquaculture production or not.
Minor Note: In Table 1, the percentages for gender have been reversed. Thank you for correcting.
Author Response
August 3, 2020
Editorial Department of “International Journal of
Environmental Research and Public Health”
Reviewer #1:
We would like to thank the reviewer for his/her comments and suggestions in relation to the work entitled “Social acceptance of aquaculture in Spain: An instrument to achieve sustainability for society”. Naturally, we have taken all these suggestions regarding content or format into consideration and implemented the corresponding amendments.
All changes corresponding to such suggestions have been highlighted in blue colour fonts in our manuscript. Such modifications have been implemented by closely following those comments:
- “This submitted paper investigates the social acceptance of aquaculture using carrying capacity as an analytical framework. Carrying capacity refers to the point at which the presence of aquaculture tends to be excessive and generate difficulties in the society. To this end, the authors conduct 803 face-to-face surveys in six coastal provinces in Spain. Their main finding suggests a high level of acceptance for aquaculture products. They conclude that aquaculture is far from its social saturation point. This submission is very interesting. It tackles in an original way an important question concerning the economy, health and the environment. This paper seems to me publishable in the journal even if I suggest a minor revision.” => We appreciate that the reviewer considers our work interesting.
- “First, it would be useful to pay more attention to the differences in responses across income categories. More specifically, I suggest that the authors give more economic insights into these differences.” => When possible, we have developed this analysis further in results and discussion (Lines 179-181, 192-200, 211-216, 250-251, 266-267, 272-273, 277-279, 287-290, 354-356). We do not know clearly if this is what the reviewer is referring to.
- “For example, is there an unequivocal link between income level and price sensitivity? How to explain the fact that "respondents with higher levels of purchasing power were the most 193 interested in the affordable price" (page 5)?” => We have added “These opinions are just the opposite for lower-income respondents (23.67%, 27.81%, 0.59% and 15.98% respectively)” in lines 195-196 to compare both positions more clearly. As explained later in this letter, we consider that the main reason for this is the level of education of the respondent. This variable has not been taken into account for this work and can be an interesting future line of research. (We suppose that the reviewer means 195. 195 refers to the total sample, not specifically to those with respondents with higher levels of income).
- “Second, is the surveyed population representative of the inhabitants of the different regions studied?” => We have worked with these provinces as a whole, due to the requirements established in the research project of which this study is part. This project appears explained in the acknowledgments of the document. If we had worked with samples stratified by provinces, the costs would have been much higher and we would have needed a larger budget.
- “Third, respond with lower income tends to be less concerned about the possible environmental damage of aquaculture. Is it due to a lack of expertise or to other factors?” => Maybe. We think that the key also lies in the correlation existing in society between the level of income and the education received. It would be necessary to deepen this question in future researches, to obtain empirical evidence in this regard. We have added this idea in the conclusions (Lines 370-375).
- “Last, for future works, it could be interesting to assess to what extent the sensitivity of the respondents differ according to the local aquaculture production or not.” => This is a good suggestion. We have added in lines 368-370: “In addition, we can also delve into whether the acceptance of aquaculture by the respondents is different towards local or foreign production”.
- “Minor Note: In Table 1, the percentages for gender have been reversed. Thank you for correcting.” => We have corrected it.

Reviewer 2 Report
The paper presents some interesting findings regarding consumer perceptions of aquaculture in Spain. Several minor improvements are suggested as followed to strengthen the paper and improve it to a publishable standard.
I would recommend clarifying the purpose of the paper and stating some clear research objectives to provide more structure to both the literature, the results, and the conclusions. At present, the purpose is stated as ‘analyse whether the carrying capacity of aquaculture in Spain supports the attainment of maximum development by taking stock of the current situation, as compared with its saturation point, or the limit beyond which it would lead to unacceptable changes in the ecosystem and social structures’. This is not what the paper does. The paper reports consumer attitudes and opinions regarding aquaculture and nothing more. The results reported include nothing about consumer views on the carrying capacity of aquaculture. The authors need to revise this and develop objectives in line with the results reported.
Results focus on reporting differences based on gender and income and yet the sampling approach included 6 different locations. So why were consumers surveyed in six different locations if no analysis was going to be done to look for differences based on location?
Author Response
August 4, 2020
Editorial Department of “International Journal of
Environmental Research and Public Health”
Reviewer #2:
We would like to thank the reviewer for his/her comments and suggestions in relation to the work entitled “Social acceptance of aquaculture in Spain: An instrument to achieve sustainability for society”. Naturally, we have taken all these suggestions regarding content or format into consideration and implemented the corresponding amendments.
All changes corresponding to such suggestions have been highlighted in green colour fonts in our manuscript. Such modifications have been implemented by closely following those comments:
- “The paper presents some interesting findings regarding consumer perceptions of aquaculture in Spain. Several minor improvements are suggested as followed to strengthen the paper and improve it to a publishable standard. ” => We would like to thank the reviewer again for revising our work.
- “I would recommend clarifying the purpose of the paper and stating some clear research objectives to provide more structure to both the literature, the results, and the conclusions. At present, the purpose is stated as ‘analyse whether the carrying capacity of aquaculture in Spain supports the attainment of maximum development by taking stock of the current situation, as compared with its saturation point, or the limit beyond which it would lead to unacceptable changes in the ecosystem and social structures’. This is not what the paper does. The paper reports consumer attitudes and opinions regarding aquaculture and nothing more. The results reported include nothing about consumer views on the carrying capacity of aquaculture. The authors need to revise this and develop objectives in line with the results reported. ” => The concept of “social carrying capacity”, explained in lines 84-94 and 328-329, is too theoretical to ask consumers directly for it, since they are usually unfamiliar with this topic. Derived from this concept, the questionnaire tried to detect if there are development possibilities for aquaculture from a social point of view, or if it generates rejection among the population for causing negative externalities in their area. For this reason, respondents were asked if they are satisfied with aquaculture, if this activity should increase in their area in the future (which would indicate that it has not reached its saturation point), if it offers healthy food, if it is environmentally friendly, if it generates quality employment, etc...
Under this approach, we have modified and developed results and discussions to adapt them to our main objective. In this way, we think that everything is clearer.
- “Results focus on reporting differences based on gender and income and yet the sampling approach included 6 different locations. So why were consumers surveyed in six different locations if no analysis was going to be done to look for differences based on location? ” => We have worked with these provinces as a whole, due to the requirements established in the research project of which this study is part. This project appears explained in the acknowledgments of the document. If we had worked with samples stratified by provinces, the costs would have been too much higher and we would have needed a larger budget. Anyway, we have included “analysis could be focused on a specific province” as a future line of research (Line 367).

Reviewer 3 Report
This article was well written and researched. This article will contribute a lot to aquaculture policy studies and consumer behavior studies. This article should be read by policy makers and NGOs.
Author Response
August 4, 2020
Editorial Department of “International Journal of
Environmental Research and Public Health”
Reviewer #3:
We would like to thank the reviewer for his/her comments in relation to the work entitled “Social acceptance of aquaculture in Spain: An instrument to achieve sustainability for society”. We appreciate that you consider our work interesting to be published.
